# Modelling Microglial Innate Immune Memory In Vitro: Understanding the Role of Aerobic Glycolysis in Innate Immune Memory

**DOI:** 10.3390/ijms24108967

**Published:** 2023-05-18

**Authors:** Morgan Towriss, Brian MacVicar, Annie Vogel Ciernia

**Affiliations:** 1Djavad Mowafaghian Centre for Brain Health, University of British Columbia, Vancouver, BC V6T 1Z3, Canada; 2Department of Biochemistry and Molecular Biology, Faculty of Medicine, University of British Columbia, Vancouver, BC V6T 1Z3, Canada; 3Department of Psychiatry, Faculty of Medicine, University of British Columbia, Vancouver, BC V6T 1Z3, Canada

**Keywords:** microglia, innate immune memory, training, tolerance, BV2, metabolism, DOHaD

## Abstract

Microglia, the resident macrophages of the central nervous system, play important roles in maintaining brain homeostasis and facilitating the brain’s innate immune responses. Following immune challenges microglia also retain immune memories, which can alter responses to secondary inflammatory challenges. Microglia have two main memory states, training and tolerance, which are associated with increased and attenuated expression of inflammatory cytokines, respectively. However, the mechanisms differentiating these two distinct states are not well understood. We investigated mechanisms underlying training versus tolerance memory paradigms in vitro in BV2 cells using B-cell-activating factor (BAFF) or bacterial lipopolysaccharide (LPS) as a priming stimulus followed by LPS as a second stimulus. BAFF followed by LPS showed enhanced responses indicative of priming, whereas LPS followed by LPS as the second stimulus caused reduced responses suggestive of tolerance. The main difference between the BAFF versus the LPS stimulus was the induction of aerobic glycolysis by LPS. Inhibiting aerobic glycolysis during the priming stimulus using sodium oxamate prevented the establishment of the tolerized memory state. In addition, tolerized microglia were unable to induce aerobic glycolysis upon LPS restimulus. Therefore, we conclude that aerobic glycolysis triggered by the first LPS stimulus was a critical step in the induction of innate immune tolerance.

## 1. Introduction

The developmental origins of health and disease (DOHaD) theory hypothesizes that environmental exposures during early life can permanently influence health and disease vulnerability in later life. This field of research has gained traction in recent years, and there is growing epidemiological and clinical evidence supporting the hypothesis [1,2]. However, there have been far fewer studies identifying potential mechanisms for mediating environmentally driven long-term impacts on cellular function and disease susceptibility [3]. One of the emerging possibilities for the mechanism is the reprogramming of metabolism in immune cells [4]. Evidence from both animal and human studies implicates the immune system in a number of disorders with known or suspected developmental origins, including schizophrenia and autism. An emerging body of evidence supports a “two-hit” hypothesis for these disorders, where an adverse early life event, such as inflammation, combines with a second adverse event in later life to impair brain development [5,6]. For example, early-life infections or brain injuries cause brain inflammation, which increases the risk of later-life brain disorders. Bacterial infections in the second and third trimester increase risk for schizophrenia [1,7,8]. Similarly, early-life infections or brain injury increase risk for earlier-onset Alzheimer’s disease (AD) with more severe cognitive decline [9,10,11,12]. How these infections lead to increased disease risk, often years to decades after the initial insult, has prompted investigation of long-term cellular reprogramming of immune cells.

Microglia, which are long-lived (4–20 years in humans [13]) resident macrophages of the brain, have become the focus of intense research to uncover how inflammation impacts the brain across the lifespan. By mimicking infection with injections of lipopolysaccharide (LPS) that produce brain inflammation, recent work has revealed that LPS can reprogram microglia to drive either enhanced (trained) [14] or repressed (tolerized) [14,15,16,17] gene expression in response to a second challenge (e.g., LPS [15,16], stroke [14] or AD plaques [14]). These immune memory effects are long-lasting (>6 months in mice) [14,15,16,17] and are hypothesized to underlie how infection and injury increase risk for subsequent neuropsychiatric and neurodegenerative disease. Several studies on microglia have shown that an initial immune challenge during development can result in a prolonged tolerized immune response. For example, Schaafsma et al. challenged pregnant females with LPS during late gestation and found that the offspring had blunted cytokine responses to subsequent immune challenge in the adult brain [18]. More recently, Hayes et al. performed a similar experiment using in utero polyI:C to mimic gestational viral infections and found prolonged blunting of microglial gene expression in response to a subsequent LPS challenge in adolescence [16]. These reprogramming events can also lead to long-term behavioural adaptations. For example, early postnatal *E. coli* infection can reprogram microglial regulation of *Il-1b* leading to hyper-induction in response to subsequent LPS exposure during adolescence [19] and a specific deficit in long-term memory only in animals receiving both “hits” (infection + LPS) [20]. Together these studies demonstrate that microglial reprogramming can occur in vivo with lasting impacts on cellular function and behaviours related to neurodevelopmental impairments.

The majority of mechanistic studies on innate immune memory have focused on monocytes and macrophages. In response to infection or vaccination, macrophages can develop a form of trained immunity that results in increased responsiveness to a subsequent immune challenge (i.e., increased responsiveness, increased production of inflammatory mediators and enhanced capacity to eliminate infection) [21]. One of the hallmarks of this form of immune memory is that it is non-specific, meaning that the initial training produces a hyper-response to a wide variety of secondary stimuli. This non-specific enhancement of immune function is thought to promote protection against future invading pathogens [22]. In comparison, tolerance may help protect the host from chronic immune activation in response to common environmental triggers such as the induction of mucosal tolerance towards the host microbiome. Either form of immune memory, when inappropriately activated, could become maladaptive, leading to chronic inflammation or autoinflammatory disease.

Early work in cultured bone marrow-derived macrophages demonstrated that LPS treatment selectively suppressed expression of pro-inflammatory genes in response to a second LPS stimulus [23]. Anti-microbial related genes did not tolerize and were equally induced by both LPS stimulations. These gene-selective effects were mediated through specific epigenetic modifications to the promoters of tolerized genes [23]. These findings have subsequently been replicated and expanded by other groups to demonstrate that epigenetic reprogramming of macrophage gene expression is a critical mechanism mediating innate immune memory [24,25,26,27]. The epigenetic regulation of innate immune memory is closely tied to metabolic reprogramming. Peripheral monocytes, like microglia, experience metabolic reprogramming when exposed to an immune agonist. For example, bacterial immune agonists, like LPS, induce a metabolic change in macrophages that is driven by increased histone acetylation and methylation which control the gene expression program [28,29]. Similar research has been conducted in microglia which revealed extensive epigenetic remodelling following LPS [15,30,31].

Microglia are known to be a highly metabolically flexible cell type, as they can rapidly adapt their energy metabolism in response to varied nutrient availability [32]. In homeostatic conditions, the switch from glycolysis to alternative metabolism mechanisms such as glutaminolysis has been shown to be essential for microglia to survey the brain parenchyma when glucose concentrations fall [33]. During an inflammatory challenge, microglia respond through complex metabolic, transcriptional and epigenomic changes that broadly result in an ‘activated’ state [29,34]. A key factor of microglial activation is that microglia undergo metabolic reprogramming in which the cells shift away from dependence on aerobic metabolism through oxidative phosphorylation and instead rely on solely glycolysis [35,36]. As a consequence of microglia switching to aerobic glycolysis, there is a rapid increase in glucose consumption and lactate production [33,35,36]. One suggested purpose of this metabolic reprogramming is that inhibition of oxidative phosphorylation results in the interruption of the tricarboxylic acid (TCA) cycle and the subsequent accumulation of TCA metabolites [21,37,38]. These metabolites are often then repurposed in the cells, supporting other functions that may be essential for microglial activation. As such, it has been identified that metabolic reprogramming is a necessary step to allow proper microglial immune activation and induction of cytokine production. However, the role of cellular metabolic shifts in establishing or controlling training and tolerance in microglia is less well characterized.

To begin to address this knowledge gap, we have developed a novel culture model of microglial training and tolerance and identified that tolerance, but not training, arises due to rapid glycolytic shifts during the initial stimulus.

## 2. Results

### 2.1. Two Hits of LPS Produce Tolerance In Vitro

#### 2.1.1. Two Hits of LPS Result in Attenuation of Inflammatory Gene Expression

Previous research has established that BV2 cells are a suitable model for examining neuroinflammation in microglia in vitro [39,40,41]; however, limited research has examined immune memory in BV2 cells. We initially aimed to model the excessively inflammatory phenotype, ‘training’, through a two-hit LPS model, as conducted previously in vivo [14]. To do so, we first treated the BV2 cells with 25 ng/mL LPS or H_2_O for 3 h, after which we removed the media and replaced them with fresh experimental media. After 24 h of culture, we treated the microglia with 25 ng/mL LPS or H_2_O for 3 h prior to lysis and gene expression analysis (Figure 1). To assess a memory phenotype, we wanted to examine if there was a statistical interaction between the first and second stimulus and if there was a difference in post hoc analysis between the “acute active” control (H_2_O LPS) and the experimental condition (LPS LPS). When examining the expression of inflammatory genes (*Il1b*, *Tnfa*, *Cxcl16*), we were surprised to observe that the expression was attenuated rather than excessive relative to the acute active control for *Il1b* (*p* < 0.0001), *Tnfa* (*p* < 0.0001) and *Cxcl16* (*p* < 0.0001) (Figure 1B, Appendix A). In accordance, two-way ANOVA revealed significant interactions for *Il1b* (F(1,63) = 295.04, *p* < 0.0001), Tnfa (F(1,52) = 187.58, *p* < 0.001) and Cxcl16 (F(1,41) = 81.03, *p* < 0.0001). Interestingly, this effect was unique to the pro-inflammatory genes; when examining the gene expression for anti-inflammatory cytokines, Tukey’s posthocs revealed that there was no difference between the experimental condition and the acute active control for Il10 (*p* = 0.8533) or Il6 (*p* = 0.1623) (Figure 1C). However, as there was residual expression as a result of the first hit, the ANOVA results found that there was a significant interaction for Il10 (F(1,41) = 13.26, *p* < 0.0001) and Il6 (F(1,42) = 551.3, *p* < 0.001). Together, this suggested that the two-hit LPS paradigm was more similar to tolerance rather than training.

#### 2.1.2. Two Hits of LPS Result in Attenuation of Microglial Phagocytosis and NO Release

Our gene expression findings indicated that two hits of LPS produced tolerance in culture. Previous research has highlighted that microglia inflammatory gene expression correlates with induction of phagocytosis both in vivo and in BV2 cells [42]. We therefore aimed to examine whether two hits of LPS produced tolerance of phagocytosis. Using a similar paradigm as for gene expression, we extended the second hit timing to 24 h to allow for quantification of cellular responses to the second hit. During the final hour of treatment, we added pHrodo *E. coli* beads to the media to assess phagocytosis by flow cytometry. Cells were gated first for size and then for singlets, and finally we assessed whether the cells contained beads or not using a ‘No bead’ control. When examining the percentage of bead-positive cells, we observed that there was a significant attenuation of phagocytosing cells between the acute active and experimental conditions (*p* = 0.0003) (Figure 1D), consistent with a tolerance of microglia phagocytosis. In accordance, there was a significant effect of the second stimulus (F(1,20) = 48.25, *p* < 0.0001) and interaction (F(1,20) = 56.63, *p* < 0.0001) but not the first stimulus (F(1,20) = 0.1477, *p* = 0.7048).

To further examine microglia function, we measured the release of NO into the media 24 h after the second hit of LPS (Figure 1E). There was a significant main effect of the first hit (F(1,20) = 6.341, *p* = 0.0204) and the second hit (F(1,20) = 78.26, *p* < 0.0001), as well as a significant interaction (F(1,20) = 79.48, *p* < 0.0001). Tukey’s corrected post hoc analysis revealed a significant attenuation of NO release when comparing the acute active condition to the experimental condition (*p* < 0.0001). There was no significant increase between the first hit control (LPS H_2_O) and the experimental condition (*p* > 0.9999). Together, these data suggest that the combined two-hit LPS paradigm results in a tolerance-like memory state in the BV2 microglia.

### 2.2. BAFF Pre-Treatment Produces Training In Vitro

#### 2.2.1. BAFF Pre-Treatment Results in Excessive Induction of Inflammatory Gene Expression upon LPS Stimulus

We aimed to model training by first changing the dose of LPS given during the first hit; however, neither a higher nor lower dose resulted in training gene expression (Appendix A). A different strategy was suggested based on another study that used B-cell-activating factor (BAFF) as an initial priming stimulus to induce training both in BV2 cells and in vivo [43]. We therefore examined BAFF’s actions as a priming stimulus in our two-hit paradigm to assess gene expression changes. BV2 microglia were treated with BAFF (10 ng/mL) or H_2_O for 3 h prior to washout and media replacement. After 24 h of culture, we treated the cells with LPS (25 ng/mL) or H_2_O for 3 h prior to lysis and gene expression analysis (Figure 2A). When comparing the acute active condition (H_2_O LPS) to the experimental condition (BAFF LPS) using Tukey’s post hocs, there was a significant enhancement of gene expression for *Il1b* (*p* < 0.0001), *Tnfa* (*p* < 0.0001) and *Cxcl16* (*p* < 0.0001) (Figure 2B, Appendix A). Two-way ANOVA indicated that there was a significant effect of the interaction for *Il1b* (F(1,28) = 7.830, *p* = 0.0092), *Tnfa* (F(1,28) = 8.845, *p* = 0.006) and *Cxcl16* (F(1,20) = 49.31, *p* < 0.0001). These results suggest that BAFF triggers a training memory state that is not solely due to the additive effects of the initial BAFF stimulus. Interestingly, analysis of anti-inflammatory gene expression again revealed no significant difference between the acute active and experimental conditions for *Il6* (*p* = 0.9963) and *Il10* (*p* = 0.7934) (Figure 2C). This supports the hypothesis that BAFF induces a training-like memory state.

#### 2.2.2. BAFF Pre-Treatment Results in Excessive Induction of Phagocytosis and NO Release

To confirm the training memory state, we examined functional assays for phagocytosis and NO release. When examining the percentage of cells phagocytosing after 1 h of bead incubation, we observed a significant interaction between the BAFF treatment and the subsequent LPS treatment (F(1,20) = 4.708, *p* = 0.0422) (Figure 2D). In addition, Tukey’s post hoc analysis showed a significant increase in the BAFF-pre-treated experimental condition compared to the acute active control (*p* = 0.0090). Similarly, when examining NO release by Griess reagent assay, we observed a significant effect of the interaction (F(1,20) = 6.228, *p* = 0.0214) and a significant change between the acute active condition and BAFF-pre-treated experimental condition (*p* = 0.0003) (Figure 2E). Together, these data demonstrate that BAFF pre-treatment results in a training memory phenotype in BV2 cells.

### 2.3. LPS but Not BAFF Results in Rapid Changes in Cellular Metabolism and Induction of Aerobic Glycolysis

Having established that a first stimulus of LPS induces tolerance, whereas a first stimulus of BAFF induces training, we next aimed to compare the acute effects of BAFF vs LPS treatment to understand how the two treatments produce distinct types of microglial memory. To do so, we treated BV2 cells with LPS (25 ng/mL), BAFF (10 ng/mL) or H_2_O for 3 h prior to lysis and gene expression analysis. At 3 h, there is a significant effect of the treatment for both inflammatory and anti-inflammatory genes. For both pro- and anti-inflammatory genes, LPS treatment resulted in a larger induction of gene expression relative to the control than BAFF treatment (Figure 3A,B, Appendix A). These data suggest that BAFF overall results in a milder gene expression response than LPS.

In microglia, an important characteristic of the response to LPS is the switch to aerobic glycolytic metabolism [33,36]. Previous research has shown that LPS-treated microglia induce aerobic glycolysis rapidly, which is required for the inflammatory activation [29,44], whereas BAFF, in addition to being an NFkB agonist, is known to stimulate the mTOR-Akt signaling pathway, promoting glycolytic metabolism more slowly [45,46]. We hypothesized that, at 24 h, BAFF may not induce aerobic glycolysis, which would support the milder gene expression response and the observed memory state (Figure 3). To test this, we examined the cellular levels of L-lactate after treatment with LPS (25 ng/mL), BAFF (10 ng/mL) or H_2_O using an ELISA-based assay. At 24 h, there was a significant effect of the treatment (F(2,15) = 88.52, *p* < 0.0001). LPS treatment resulted in a significant increase in L-lactate (*p* < 0.0001), while in contrast there was no significant change in the levels of L-lactate in the BAFF-treated cells versus the H_2_O control (*p* = 0.1796) (Figure 3C). This supports our hypothesis that BAFF fails to induce a metabolic shift to aerobic glycolysis at 24 h. We therefore hypothesized that the rapid induction of aerobic glycolysis was required to induce the tolerance-like phenotype observed following LPS pre-treatment.

### 2.4. Inhibition of LPS-Induced Aerobic Glycolysis Impairs Establishment of Microglial Memory

#### 2.4.1. Sodium Oxamate Attenuates LPS-Induced Gene Expression and Blocks Lactate Production

To test the hypothesis, we next aimed to use a drug that would block aerobic glycolysis without inhibiting normal aerobic metabolism. The purpose of fermentation in a cell is to replenish the cytosolic NAD+. In aerobic conditions, this can occur through oxidative phosphorylation; however, fermentation is required in anaerobic or high-energy-demand conditions to replenish NAD+. To test the requirement of aerobic glycolysis for LPS-induced tolerance in BV2 cells, we utilized sodium oxamate to block fermentation through inhibition of L-lactate dehydrogenase (Figure 4A). We treated BV2 cells with sodium oxamate (10 mM) concurrently with LPS (25 ng/mL) and examined the change in gene expression at 3 h. As expected, sodium oxamate resulted in an attenuation of LPS-induced gene expression for *Il1b*, *Tnfa*, *Il10* and *Il6* (Figure 4B,C, Appendix A). When examining the change in L-lactate production, we observed that there was a significant effect of sodium oxamate (F(1,20) = 157.1, *p* < 0.0001) and the LPS stimulus (F(1,20) = 148.2, *p* < 0.0001), as well as a significant interaction (F(1,20) = 85.14, *p* < 0.0001). Tukey’s post hoc analysis revealed that there was no significant change between the vehicle H_2_O control and the sodium oxamate LPS condition (*p* = 0.9940), confirming that sodium oxamate successfully blocked the LPS-induced production of L-lactate and thus aerobic glycolysis.

#### 2.4.2. Treatment of Sodium Oxamate during the First Stimulus Blocks Establishment of the Tolerance Memory State Gene Expression but Not the Training Memory State

After confirming the efficacy of sodium oxamate to block LPS-induced aerobic glycolysis, we next sought to examine the impact of blocking aerobic glycolysis during the establishment of LPS-induced tolerance memory. We treated BV2 cells concurrently with sodium oxamate (10 mM) or vehicle and LPS (25 ng/mL) or H_2_O for 3 h followed by washout and replacement with experimental media. BV2 cells were then treated with 25 ng/mL LPS or H_2_O for 3 h prior to lysis and analysis (Figure 5A). To assess if sodium oxamate impacted the establishment of the tolerance memory state, we ran a three-way ANOVA with drug treatment, first hit and second hit as the three factors. To observe that there was a significant change in the establishment of the tolerance memory state, we were looking for a significant interaction between the three variables and a change in the induction of gene expression between the vehicle control two-hit condition (veh LPS LPS) and the sodium oxamate two-hit condition (SO LPS LPS). At 3 h after the second stimulus, for *Il1b* there is a significant interaction between sodium oxamate treatment, the first hit and the second hit (F(1,61) = 12.37, *p* = 0.0008) (Figure 5B, Appendix A). Tukey’s post hoc analysis revealed that there is a significant difference between the vehicle-treated LPS LPS condition and the sodium oxamate-treated LPS LPS condition (*p* < 0.0001). As such, while there is a significant difference between H_2_O LPS and LPS LPS for the vehicle condition (*p* < 0.0001), there is no significant difference observed for the sodium oxamate-treated H_2_O LPS and LPS LPS conditions (*p* = 0.8105), suggesting that the production of L-lactate and the switch to aerobic glycolysis is required for LPS-induced tolerance memory in BV2 cells. Similarly, for *Tnfa* there is a significant interaction (F(1,60) = 32.58, *p* < 0.0001) between sodium oxamate treatment, the first hit and the second hit (Figure 5C). Tukey’s post hoc analysis revealed a significant difference between the vehicle-treated LPS LPS condition and the sodium oxamate-treated LPS LPS condition (*p* < 0.0001) and no difference between the sodium oxamate-treated H_2_O LPS and LPS LPS conditions (*p* > 0.9999). Finally, for *Il10* there was no significant effect of the drug treatment (F(1,61) = 3.501, *p* = 0.0661) or interaction between the drug treatment, first hit and second hit (F(1,61) = 0.02728, *p* = 0.8694) (Figure 5C). Likewise, Tukey’s post hoc analysis showed no difference between the sodium oxamate-treated LPS LPS and the vehicle-treated LPS LPS conditions (*p* = 0.9868).

We next assessed the impact of inhibition of aerobic glycolysis through sodium oxamate on the training gene expression state. We treated BV2 cells concurrently with sodium oxamate (10 mM) or vehicle and BAFF (10 ng/mL) or H_2_O for the first stimulus, followed by the second stimulus as previously described (Appendix A). As BAFF does not rapidly induce aerobic glycolysis, we did not expect that sodium oxamate would impact the training gene expression. When examining pro-inflammatory cytokine gene expression, sodium oxamate had no effect on the gene expression induction in the memory state (Appendix A). This further supports that the BAFF-induced training memory phenotype occurs by a distinct mechanism from LPS-induced tolerance and does not require rapid shifts to aerobic glycolysis. 

#### 2.4.3. Sodium Oxamate Rescues Attenuation of Phagocytosis and NO Production

To examine how inhibition of aerobic glycolysis in the context of LPS-induced tolerance impacts microglial function, we assessed if concurrently treating the cells with sodium oxamate during the first LPS stimulus would rescue the tolerance-like functional assays. We treated BV2 cells with sodium oxamate (10 mM) or vehicle and LPS (25 ng/mL) or H_2_O for 3 h followed by washout and replacement with experimental media. We then treated the cells with LPS (25 ng/mL) or H_2_O for 24 h prior to functional assessment (Figure 6A). When assessing phagocytosis, we found that there was a significant interaction between the first hit, second hit and sodium oxamate treatment (F(1,64) = 12.86, *p* = 0.0006), suggesting that sodium oxamate impacted the interaction between the first and second hit observed previously (Figure 6B, Appendix A). Tukey’s post hoc analysis revealed that there is a significant difference between the vehicle-treated LPS LPS condition and the sodium oxamate-treated LPS LPS condition (*p* < 0.0001). In addition, the significant difference observed between veh LPS and LPS LPS (*p* < 0.0001) was no longer present after treatment with sodium oxamate (*p* = 0.1533). For the NO production assay, similar to the phagocytosis assay, there were significant interactions between the first hit, second hit and sodium oxamate treatment (F(1,40) = 4.862 (*p* = 0.0333), suggesting that sodium oxamate impacts the interaction observed between the first and second hit (Figure 6C). Tukey’s post hoc analysis shows that there is a significant difference between the vehicle LPS LPS condition and the sodium oxamate LPS LPS condition (*p* < 0.0001). However, unlike the phagocytosis assay, this was an incomplete rescue, as there is still a significant attenuation of NO production in the SO LPS LPS condition relative to the SO veh LPS condition (*p* = 0.0034). Altogether, these data suggest that treatment with sodium oxamate during the first LPS stimulus impairs the establishment of the tolerance-like memory state, whereas treatment with sodium oxamate during a BAFF stimulus does not. Therefore, we proposed that the glycolytic flux during the first stimulus is critical for the long-term inhibition of the cellular pro-inflammatory state. We then hypothesized that the tolerance phenotype could occur in part as a result of attenuation of aerobic glycolysis.

### 2.5. Tolerized Microglial Metabolic State Is Rescued by Sodium Oxamate Treatment during the First Hit

#### 2.5.1. Two Hit of LPS Block Aerobic Glycolysis, whereas BAFF Pre-Treatment Enhances Aerobic Glycolysis

We hypothesized that in the tolerized state, aerobic glycolysis would be attenuated, similar to the suppressed inflammatory gene expression. To examine the cell metabolic state, we performed L-lactate analysis 24 h after the second LPS stimulus. As hypothesized, the lactate data showed a significant interaction between the first and second hit (F(1,20) = 100.7, *p* < 0.0001). Tukey’s post hoc analysis revealed a significant attenuation of lactate production for the LPS LPS condition compared to the acute active condition (veh LPS) (*p* < 0.0001) (Figure 7A). For the BAFF first treatment (Figure 7B, Appendix A), there was a significant interaction between the first and second stimulus (F(1,20) = 8.681, *p* = 0.0080). Tukey’s post hoc analysis revealed a significant enhancement of lactate production by BAFF pre-treatment compared to the acute active condition (veh LPS) (*p* = 0.003). Together, these findings support that enhanced lactate production occurs under conditions that also produce enhanced pro-inflammatory gene expression and that repressed lactate production occurs under conditions that also suppress inflammatory gene expression.

#### 2.5.2. Sodium Oxamate Treatment Prevents the Tolerance-Induced Metabolic State

We next aimed to assess whether the sodium oxamate-mediated de-repression of gene expression would also reverse the suppression of aerobic glycolysis. Therefore, using the same paradigm (Figure 6A), we assessed the lactate production following sodium oxamate treatment (Figure 7C). Three-way ANOVA analysis revealed a significant interaction of the first hit, second hit and sodium oxamate treatment (F(1,40) = 26.46, *p* < 0.0001), suggesting that sodium oxamate treatment impacted the observed interaction between the first and second hit. Tukey’s post hoc comparisons showed this was true, as there was a significant difference in the vehicle and sodium oxamate-treated LPS LPS conditions (veh LPS LPS vs. SO LPS LPS, *p* < 0.0001), and there was no longer a significant difference between the SO-treated acute active condition and the sodium oxamate-treated LPS LPS condition (SO veh LPS vs. SO LPS LPS, *p* > 0.9999). Altogether, these data suggest that there is a metabolic mediated shut-off mechanism occurring following the first LPS stimulus which controls tolerance gene expression and functional states. This is supported by inhibition of metabolic flux through sodium oxamate preventing the establishment of tolerized gene expression, phagocytosis, NO release and lactate production.

## 3. Discussion

In the present study, we showed that BV2 cells can be used as a model to study innate immune memory in microglia in vitro. Using this model, we established that trained and tolerized microglia can be established in the same paradigm using different initial treatments. After establishing that LPS produces a tolerance state and BAFF produces a trained state, we examined the difference between the two stimuli and observed that LPS rapidly induced aerobic glycolysis, whereas BAFF did not. We hypothesized that the switch to an aerobic glycolysis metabolism during the first LPS stimulus was responsible for tolerance of inflammatory gene expression and showed this to be true using sodium oxamate as a transient inhibitor of glycolytic flux during the initial LPS stimulus. Inhibition of glycolytic flux prevented LPS-induced tolerance but did not impact BAFF-induced training in a parallel experiment. As such, we propose that in vitro microglial tolerance, but not training, occurs due to the rapid initiation of aerobic glycolysis during the first stimulus, which results in long-term suppression of aerobic glycolysis and inflammatory gene expression.

The findings from our in vitro model are consistent with previous work on in vivo microglia that found LPS-induced tolerance to a secondary stimulus [14,15,16]. In in vivo models, these effects are long-lasting (1–6 month) [14,15,16], suggesting stable cellular reprogramming. Whether this reprogramming occurs directly through long-term suppression of aerobic glycolysis or through other mechanisms has not been tested. Long-term epigenetic changes and altered gene expression have been observed [14,15,16], but again not causally tested mechanistically. Our findings support a model in which metabolic shifts are the primary driver of the formation of long-term innate immune memories. This is particularly important for microglia, which are long-lived cells in the brain [13], potentially allowing for long-term reprogramming events to impact disease risk across the lifespan. For example, infections during pregnancy increase the risk of the child subsequently developing a neurodevelopmental or neuropsychiatric disorder in later life. In animal maternal immune activation (MIA) during gestation models, viral or bacterial infection or treatment with chemicals that mimic infection through immune activation (e.g., LPS or polyI:C) can drive neurodevelopmental abnormalities and abnormal behaviours [47]. Alterations in microglial functions are thought to contribute to the long-term impacts of MIA. For example, maternal allergic asthma is a risk factor for autism spectrum disorder, and in a mouse model, simulated asthma attacks during pregnancy activate the dam’s immune response, increase cytokine levels in the fetal brain and resulting in offspring with autism-like behavioural deficits [48,49,50,51,52]. In this model, microglia are also impacted by alterations in gene expression and DNA methylation into adolescence [53], suggesting that epigenetic remodelling in response to MIA may result in a form of maladaptive innate immune memory. In support of this finding, in utero polyI:C exposure to mimic a viral infection produced long-term blunting of microglial activity similar to tolerance [16]. When challenged as adults with LPS, microglia from mice that received polyI:C in utero showed suppression of pro-inflammatory gene expression, decreased phagocytosis activity and impaired neuronal circuity [16]. These findings are consistent with our LPS model of tolerance and suggest a potential metabolic link between early-life inflammation and later-life alterations in microglial responses that can be explored in future in vivo studies.

One of the major microglial functions disrupted in MIA models is the phagocytosis of neuronal synapses during key developmental windows of synapse pruning. During normal development, microglia engulf and remove synapses to fine-tune developing neuronal circuits [54]. Disruptions in microglial activity can alter pruning activity, resulting in abnormal neuronal circuits [55,56]. For example, combining exposure to an environmental pollutant with maternal stress results in male offspring with hypo-active microglia that fail to developmentally prune synapses, resulting in abnormal social behaviours [57]. Other early-life perturbations can also impact microglial pruning, with negative impacts on behaviour including early-life stress [58,59], use of drugs of abuse [60] and MIA [61,62]. Often, microglia-driven deficits are either exacerbated or require a second challenge in adolescence to fully manifest. For example, LPS given at postnatal day 14 exacerbated the impacts of stress given during adolescence, resulting in microglial over-pruning of excitatory synapses and depressive-like behaviours [63]. Given that both hyper- and hypo-responsive microglia have been observed in these models, we propose that our simplified culture model can capture both aspects of early-life microglial reprogramming. Our LPS-induced tolerance model suppressed microglial phagocytosis, similar to observations of synaptic engulfment in vivo. In comparison, we found increases in phagocytosis with BAFF as the first hit, together allowing us to examine potential differentiating mechanisms driving differential microglial responses in vivo. Our model also will allow for rapid high-throughput screening of target genes, metabolic regulators and other factors that can then be tested in these more complex and time-consuming in vivo systems.

Innate immune tolerance arises in the periphery following a strong immune response as a protective mechanism against tissue damage which may arise from chronic stimulus [28,64]. However, while this is typically thought of as protective, excessive tolerance is thought to result in immunoparalysis and an inability to launch a strong response to inflammatory challenge. Bacterial LPS is known to induce tolerance in peripheral monocytes in a dose-dependent manner [28,64,65]. This is consistent with findings in microglia which show chronic administration of peripheral LPS induces tolerance [14]. The data we present agree with this, as direct administration of LPS to BV2 cells would be potentially comparable to a high dose in vivo. One previous study found that ultra-low-dose administration of LPS (1 fg/mL) did not induce tolerance but rather training, suggesting that the dose effect found in vivo might also persist in vitro [66]. In peripheral monocytes, it has been suggested that the observed LPS dose effect is dependent on the initiation of mTOR signalling at low doses and AMPK signalling at high doses, both of which are master regulators of cellular metabolism [28,67]. Our work is consistent with these findings, as we found that alterations in cellular metabolism can alter memory formation. We suggest that aerobic glycolysis is a necessary factor to produce cellular tolerance, potentially through a number of signalling pathways that remain to be elucidated in vivo in microglia.

Growing bodies of evidence suggest that trained microglia contribute to neuroinflammatory disease pathology [14]. Under inflammatory conditions microglia shift from oxidative phosphorylation metabolism to aerobic glycolysis (Warburg effect); however, this effect is thought to be temporary and to possibly drive long-term suppression of microglia function [68]. Distinctly, previous research has suggested that the underlying cause of trained memory is the prolonged enhancement of aerobic glycolysis that enables a more robust and rapid response to future stimuli [46,69]. For example, peripheral monocytes have previously been shown to have epigenetically driven enhancements in the mTOR–HIF1a pathway following an immune stimulus that produces a long-term enhancement of the metabolic state and enhanced immune activation [69]. Likewise, BAFF treatment in microglia has been shown to result in a delayed long-term enhancement of glycolysis [45,46]. Similarly, previous studies have suggested that the trained memory state is associated with a glycolytic metabolism switch [14,68]. Our BAFF training model in BV2 cells is consistent with these findings, as BAFF pre-treatment results in increased lactate production in response to LPS. It would be interesting to assess if LPS given at longer time points after BAFF treatment would produce the enhancement of aerobic glycolysis. Given that excessive metabolic enhancement is a hallmark of chronic neuroinflammatory disease states [14,68], understanding how prolonged metabolic enhancement is associated with trained immunity may enable novel therapeutic development strategies for neuroinflammatory diseases.

Using the in vitro model presented, we were able to examine and manipulate cellular metabolism, which enabled the findings from this study. However, while BV2 cells are a good model for microglia, previous research has established that they are transcriptionally distinct from microglia in the brain [70,71]. One limitation of our model was that we only waited 24 h between immune stimuli, which means that the impacts from the first stimulus may not have fully resolved when the second stimulus was given. One consequence of this is that it is difficult to examine whether protein level changes of secreted immune signaling molecules in the media occur as a result of the first immune challenge or the second. We therefore examined RNA levels within the cells and assumed a degree of RNA-to-protein correlation. However, this may not be a correct assumption in all cases, and therefore further research performed with animal models should examine protein levels in addition to RNA levels. In addition, as microglia in the brain do not exist in isolation, it is possible that their cellular metabolism may be altered by the presence of other metabolites from neurons and astrocytes [72,73,74]. As a result, any findings from this model should be examined further in vivo. One strength of our model is that it allows for transient drug treatment specifically of microglia. This type of selective pharmacological manipulation is currently not possible in vivo. Consequently, our in vitro model can provide an opportunity for researchers to screen signalling pathways that might be involved in innate immune memory prior to investing in cell type-specific genetic mouse models for follow-up in vivo studies. Finally, previous research has established that sex and developmental age are important factors for microglia function and immune response [75]; as BV2 cells are derived from a female neonate primary microglia culture, there is no ability to examine sex or age effects on memory with this model.

In summary, we present findings that alterations in cellular metabolism are required to drive innate immune memory formation in microglia. The present findings suggest an important role of the switch to aerobic glycolysis; however, further studies are required to determine the signalling pathways and epigenetic mechanisms involved in the metabolic changes and inflammatory gene expression control. Future directions for this research include further examining the metabolic pathways at play both in establishing and maintaining the training and tolerance models. We also aim to examine the metabolic findings further using in vivo immune memory models. Previous research has begun to suggest a role of cellular metabolism in immune memory, with RNA sequencing revealing differential metabolism-linked genes [14,16]. However, future work is needed to more thoroughly understand the cellular metabolism of trained and tolerized microglia in vivo and understand whether the pattern of differential induction of aerobic glycolysis drives training and tolerance.

## 4. Materials and Methods

### 4.1. BV2 Microglial Culture + Treatments

BV2 cells from ATCC were grown in DMEM F12 (ThermoFisher #11320033, Waltham, MA, USA), 10% FBS (ThermoFisher #12484028, Waltham, MA, USA), 1× GlutaMAX (ThermoFisher #35050061, Waltham, MA, USA) and 1× penicillin–streptomycin (ThermoFisher #15140122, Waltham, MA, USA) at 37 °C and 5% CO_2_. Following growth, cells were passaged with 0.25% trypsin–EDTA (ThermoFisher #25200072, Waltham, MA, USA) and plated in reduced serum media for experiments (DMEM F12, 2% FBS, 1× Glut, 1× penicillin–streptomycin). BV2 microglia were treated with lipopolysaccharides (LPS) from *Escherichia coli* (SigmaAldrich #L5418, St. Louis, MI, USA), recombinant mouse B-cell activating factor (BAFF) (R&D Systems, #8876-BF, Minneapolis, MN, USA) or sodium oxamate (Alfa Aesar #A16532-06, Haverhill, MA, USA). LPS was serially diluted in water to a stock concentration of 10 μg/mL and then diluted in reduced serum media to 20x the final well concentration. LPS concentrations ranged from 0.1–100 ng/mL. BAFF was resuspended in water according to manufacturer’s instruction and stored at a stock concentration of 10 μg/mL. BAFF stock was then diluted in reduced serum media to a 20× final well concentration of 10 ng/mL. Sodium oxamate was resuspended in reduced serum media at a final concentration of 20× the final well concentration of 10 mM. Cells were treated for 3 or 24 h prior to washout or lysis. During washout, cells were centrifuged at 500× *g* for 8 min and supernatant aspirated and replaced with reduced serum media, after which the plates were returned to incubate at 37 °C and 5% CO_2_.

### 4.2. RNA-Extraction and RT-qPCR Analysis

BV2 cells were lysed in RNA lysis buffer (Zymo Research Quick-RNA Microprep kit) and rapidly frozen using dry ice. Samples were stored at −80 °C until RNA extraction. The RNA extraction was performed using a Zymo Research Quick-RNA Microprep kit (Zymo Research #R1051, Irvine, CA, USA) and the concentration determined via Nanodrop. cDNA was synthesized from a 500 ng RNA (25 ng/μL) input using LunaScript^®^ RT-SuperMix kit (New England Biolabs #E3010, Ipswich, MA, USA) and diluted to a total cDNA concentration of 10 ng/μL using nuclease-free water. RT-qPCR reactions were performed with a Luna^®^ Universal qPCR Master Mix (New England Biolabs #M3003, Ipswich, MA, USA) with 0.25 µM per primer per reaction. Primer melt curves were evaluated to confirm the presence of a single amplification product of the predicted size. Primer efficiency was validated by a standard curve. Analysis was performed with the ddCT method, with all genes normalized to the housekeeping gene (HPRT) and the vehicle controls within the experimental replicate. Primer sequences are listed in Table 1.

### 4.3. pHrodo E. coli Phagocytosis Assay Quantified by Flow Cytometry

BV2 microglial phagocytosis was assessed via flow cytometry by assaying engulfment of pHrodo Red *E. coli* BioParticles Conjugate for Phagocytosis (ThermoFisher Scientific #P35361, Waltham, MA, USA). pHrodo *E. coli* bioparticles were diluted in media to a stock concentration of 1 mg/mL and stored at −20 °C for up to 6 months. Bioparticle stock was diluted in media to 20× the final concentration of the well (1:500 stock, 2 μg/mL). The 20× bioparticle spike was added to each well during the last hour of drug/stimulus treatment. After 1 h, cells were centrifuged at 500× *g* for 5 min to remove the media and washed with 1× HBSS. Cells were resuspended in 1% PFA and incubated at room temperature for 30 min or at 4 °C overnight. After fixation, cells were washed twice in 1× HBSS and resuspended in 200 μL of FACS buffer for analysis. Cells were analyzed on the Beckman Coulter CytoFLEX (Brea, CA, USA). Compensations were performed and the voltage standardized across experiments using the median fluorescent intensity of fluorescent rainbow beads. Events were gated first for cell size on SSC-A vs. FSC-H and then for singlets on FSC-H vs. FSC-W; finally, bead-positive cells were assessed on a phycoerythrin (PE) red-channel (Y585) signal to detect cells with bead engulfment. Controls used for gating were cells that did not receive bead treatment. FlowJo was used to assess the percentage of phagocytic-positive cells gated on the no-bead control on Y585 and the median fluorescent intensity (MFI) of the positive population (Y585+).

### 4.4. Griess Reagent Assay

The Griess reagent assay (ThermoFisher #G7921, Waltham, MA, USA) was used to quantify media-released nitrite concentrations in BV2 wells. For this experiment, experimental cells were incubated in reduced serum media made from DMEMF12 without phenol red to minimize background signals. A standard curve was prepared from nitrite-containing samples and used to determine sample concentrations.

### 4.5. Lactate Assay

The L-lactate assay (Abcam #ab65331, Cambridge, UK) was performed according to manufacturer’s specifications with perchloric acid deprotonation. A standard curve was prepared from lactate-containing samples to determine the sample concentrations. BV2 cell lysates were serially diluted (1:5, 1:10) to ensure that the range of the lactate readout was within the maximum and minimum range of the standard curve for all the treatments.

### 4.6. Statistical Analysis

In the instances of only one experimental variable, ordinary one-way ANOVAs were run comparing the mean of each treatment to the control. Data were tested for normalcy using the Shapiro–Wilk test. For experiments with only two variables, a two-way ANOVA was run comparing the mean of each treatment to the mean of the vehicle controls. Tukey’s post hoc comparisons were run for individual treatment comparisons. Data were tested for normalcy using the Shapiro–Wilk test. Finally, for experiments with three treatment variables, three-way ANOVA was run for the treatment comparisons to fit a full effect model (A, B, C, A × B, A × C, B × C, A × B × C). Tukey’s post hoc comparisons were run for individual treatment comparisons. Data were tested for normalcy using the Shapiro–Wilk test and for homoscedasticity using Spearman’s test for heteroscedasticity or the Brown–Forsythe test. All data passed the normalcy and homoscedasticity tests.

## Figures and Tables

**Figure 1 ijms-24-08967-f001:**
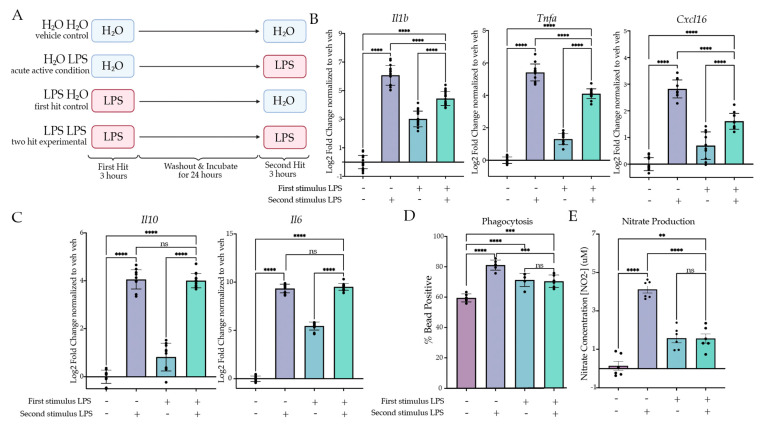
Two LPS treatments induce tolerance in culture. (**A**) BV2 cells were treated with H_2_O or Lipopolysaccharide (LPS) (25 ng/mL) for 3 h followed by washout. After 24 h of incubation, the cells were treated with H_2_O or LPS (25 ng/mL) for 3 h or 24 h prior to analysis. Created with BioRender.com. (**B**) RT qPCR assessment of gene expression of *Il1b*, *Tnfa* and *Cxcl16* in BV2 cells treated with two stimuli of H_2_O or LPS. Inflammatory gene expression assessed 3 h after the second hit. *N* = 4–6 independent experiments, 2–4 replicates per experiment. (**C**) RT qPCR assessment gene expression of *Il6* and *Il10* in BV2 cells treated with two stimuli of H_2_O or LPS. Anti-inflammatory gene expression assessed 3 h after the second hit. *N* = 4 independent experiments, 2–4 replicates per experiment. (**D**) Phagocytosis of pH-rhodo *E. coli*-tagged beads assessed at 24 h after second stimulus of LPS or H_2_O. *n* = 3 independent experiments, 2 replicates per experiment. (**E**) Release of NO measured 24 h after second LPS treatment, measured through Griess reagent assay. *N* = 3 independent experiments, 2 replicates per experiment. qPCR shown as bar graph of Log_2_(Fold Change) + SEM Tukey’s post hoc significances denoted by (ns *p* > 0.05, ** *p* < 0.01, *** *p* < 0.001, **** *p* < 0.0001). All statistics are reported in Appendix A.

**Figure 2 ijms-24-08967-f002:**
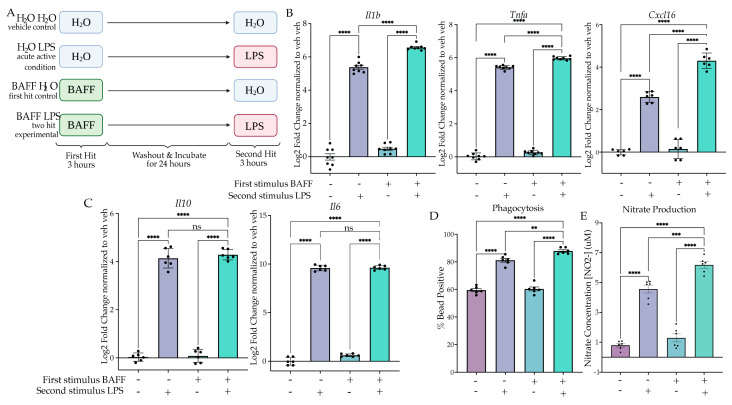
BAFF pre-treatment produces training in culture. (**A**) BV2 cells were treated with H_2_O or B-cell-activating factor (BAFF) (10 ng/mL) for 3 h followed by washout. After 24 h of incubation, the cells were treated with H_2_O or Lipopolysaccharide (LPS) (25 ng/mL) for 3 h or 24 h prior to analysis. Created with BioRender.com. (**B**) RT qPCR assessment of inflammatory gene expression of *Il1b*, *Tnfa* and *Cxcl16* in BV2 treated with a priming stimulus of BAFF or H_2_O followed by a second stimulus of H_2_O or LPS. *n* = 3 independent experiments, 2–3 replicates per experiment. (**C**) RT qPCR assessment of anti-inflammatory gene expression of *Il10* and *Il6* in BV2 treated with a priming stimulus of BAFF or H_2_O followed by a second LPS stimulus. *n* = 3 independent experiments, 2 replicates per experiment. (**D**) Phagocytosis of pH-rodo *E. coli*-tagged beads assessed at 24 h after second stimulus of LPS or H_2_O. *n* = 3 independent experiments, 2 replicates per experiment. (**E**) Release of NO measured 24 h after second LPS treatment, measured through Griess reagent assay. *n* = 3 independent experiments, 2 replicates per experiment. qPCR shown as bar graph of Log_2_(Fold Change) + SEM Tukey’s post hoc significances denoted by (ns *p* > 0.05, ** *p* < 0.002, *** *p* < 0.0002, **** *p* < 0.0001). All statistics are reported in Appendix A.

**Figure 3 ijms-24-08967-f003:**
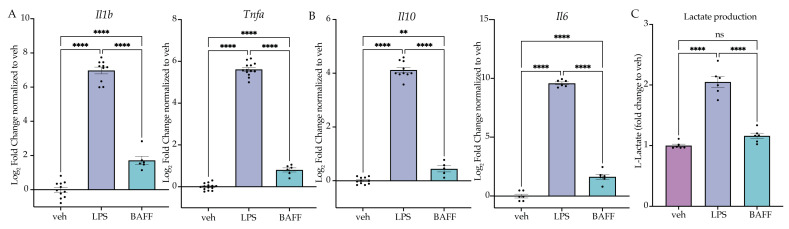
LPS but not BAFF induces rapid induction of lactate production. (**A**) BV2 cells are treated with Lipopolysaccharide (LPS) (25 ng/mL) or B-cell-activating factor (BAFF) (10 ng/mL) for 3 h prior to analysis. RT-qPCR analysis of inflammatory cytokines for inflammatory genes. *n* = 3–4 independent experiments, 2–4 replicates per experiment. (**B**) BV2 cells are treated with LPS (25 ng/mL) or BAFF (10 ng/mL) for 3 h prior to analysis. RT-qPCR analysis of anti-inflammatory cytokines Il10 and Il6. *n* = 3–4 independent experiments, 2–4 replicates per experiment. (**C**) Intracellular L-lactate in BV2 cells treated with LPS (25 ng/mL) or BAFF (10 ng/mL) recorded 24 h after treatment. *n* = 3 independent experiments, 2 replicates per experiment. qPCR shown as bar graph of Log_2_(Fold Change) + SEM Tukey’s post hoc significances denoted by (ns *p* > 0.05, ** *p* < 0.01, **** *p* < 0.0001). All statistics are reported in Appendix A.

**Figure 4 ijms-24-08967-f004:**
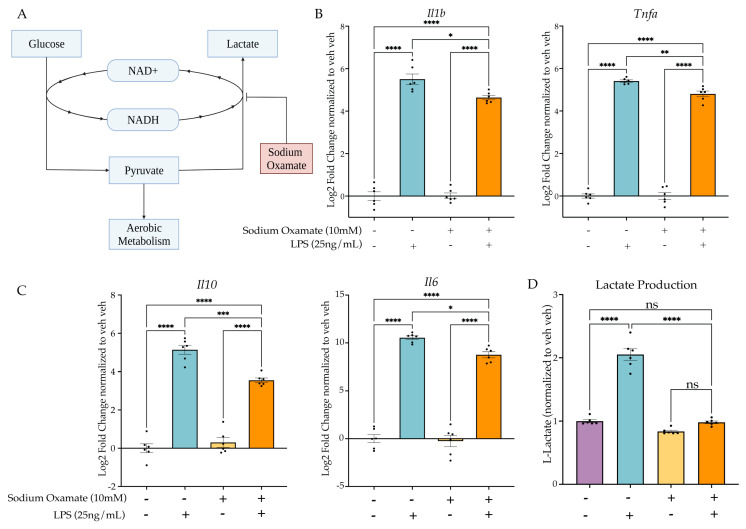
Sodium oxamate blocks lactate production and attenuates inflammatory gene expression. (**A**) Sodium oxamate blocks aerobic glycolysis by inhibiting fermentation and thus further glucose consumption due to NAD+ availability. Created with BioRender.com. (**B**) BV2 cells are treated with sodium oxamate (10 mM) or vehicle and Lipopolysaccharide (LPS) (25 ng/mL) or H_2_O for 3 h prior to lysis and analysis. RT-qPCR of inflammatory gene expression for *Il1b* and *Tnfa*. (**C**) RT-qPCR of anti-inflammatory gene expression for *Il10* and *Il6*. (**D**) Intracellular L-lactate in BV2 cells treated with LPS (25 ng/mL) or BAFF (10 ng/mL) recorded 24 h after treatment. *n* = 3 independent experiments, 2 replicates per experiment. qPCR shown as bar graph of Log_2_(Fold Change) + SEM Tukey’s post hoc significances denoted by (ns *p* > 0.05, * *p* < 0.05, ** *p* < 0.01, *** *p* < 0.001, **** *p* < 0.0001). All statistics are reported in Appendix A.

**Figure 5 ijms-24-08967-f005:**
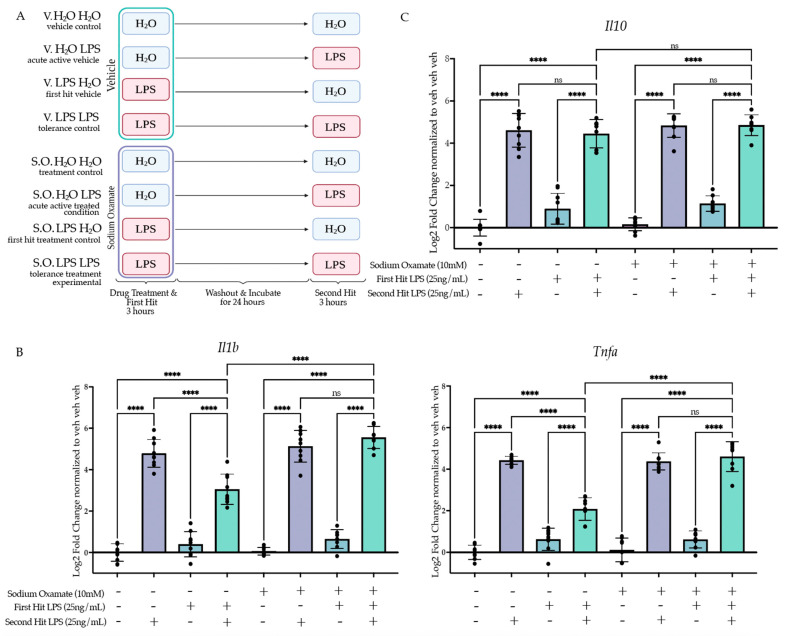
Sodium oxamate treatment during the first stimulus rescues the blunted gene expression and function outcomes of BV2 cells experiencing LPS-induced tolerance. (**A**) BV2 cells were treated with sodium oxamate (10 mM) or vehicle concurrently with H_2_O or Lipopolysaccharide (LPS) (25 ng/mL) for 3 h followed by washout. After 24 h of incubation, the cells were treated with H_2_O or LPS (25 ng/mL) for 3 h prior to analysis. Created with BioRender.com. (**B**) RT-qPCR analysis of gene expression of pro-inflammatory cytokine *Il1b* and *Tnfa* assessed 3 h after the second stimulus of LPS or H_2_O. *n* = 3 independent experiments, 2–3 replicates per experiment. (**C**) RT-qPCR analysis of gene expression of anti-inflammatory cytokine *Il10* assessed 3 h after the second stimulus of LPS or H_2_O. *n* = 3 independent experiments, 2–3 replicates per experiment. qPCR shown as bar graph of Log_2_(Fold Change) + SEM Tukey’s post hoc significances denoted by (ns *p* > 0.05, **** *p* < 0.0001). All statistics are reported in Appendix A.

**Figure 6 ijms-24-08967-f006:**
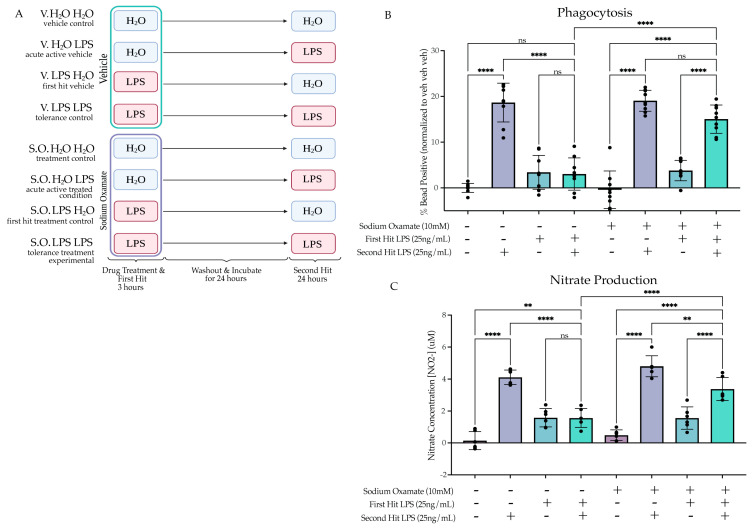
Sodium oxamate treatment rescues the tolerized induction of phagocytosis and nitrate production. (**A**) BV2 cells were treated with sodium oxamate (10 mM) or vehicle concurrently with H_2_O or Lipopolysaccharide (LPS) (25 ng/mL) for 3 h followed by washout. After 24 h of incubation, the cells were treated with H_2_O or LPS (25 ng/mL) for 24 h prior to analysis. Created with BioRender.com. (**B**) Phagocytosis of pHrodo *E. coli*-tagged beads assessed at 24 h after second stimulus of LPS or H_2_O. Measured as the percentage of cells fluorescing the beads, normalized within experimental replicate to vehicle H_2_O H_2_O. *n* = 3 independent experiments, 2 replicates per experiment. (**C**) Release of NO measured 24 h after second LPS treatment, measured through Griess reagent assay. *n* = 3 independent experiments, 2 replicates per experiment. Graphs shown as bar graph + SEM Tukey’s post hoc significances denoted by (ns *p* > 0.05, ** *p* < 0.01, **** *p* < 0.0001). All statistics are reported in Appendix A.

**Figure 7 ijms-24-08967-f007:**
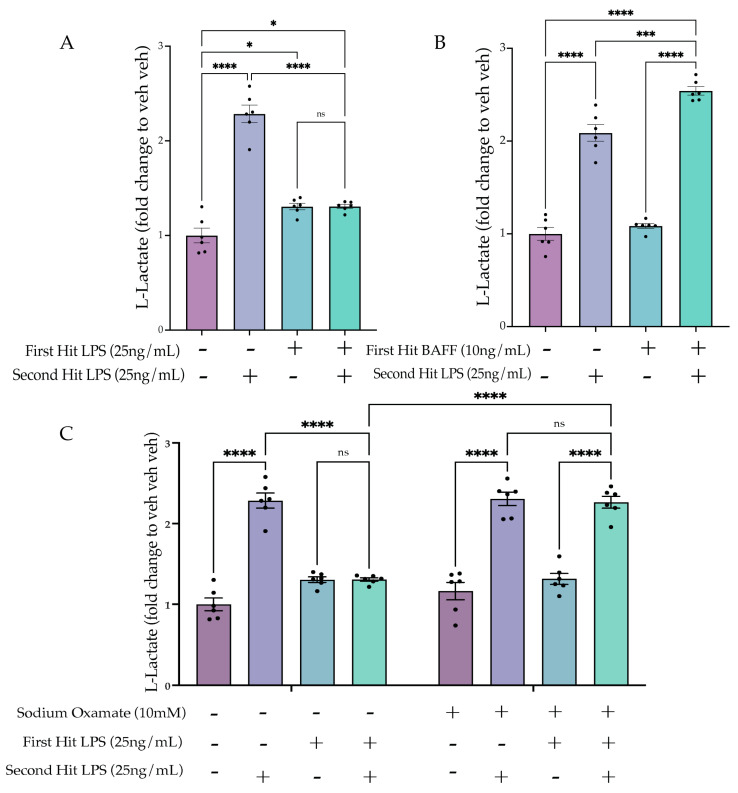
Sodium oxamate prevents tolerized aerobic metabolism. (**A**) BV2 cells are treated with Lipopolysaccharide (LPS) (25 ng/mL) or H_2_O for 3 h prior to washout; 24 h after, the cells are treated with LPS (25 ng/mL) or H_2_O for 24 h prior to analysis. L-lactate levels are depicted as a fold change relative to the H_2_O H_2_O control. *n* = 3 independent experiments, 2 replicates per experiment. (**B**) BV2 cells are treated with B-cell-activating factor (BAFF) (10 ng/mL) or H_2_O for 3 h prior to washout; 24 h afterward, the cells are treated with LPS (25 ng/mL) or H_2_O for 24 h prior to analysis. L-lactate levels are depicted as a fold change relative to the H_2_O H_2_O control. *n* = 3 independent experiments, 2 replicates per experiment. (**C**) BV2 cells are treated with sodium oxamate (10 mM) or vehicle and LPS (25 ng/mL) or H_2_O for 3 h prior to washout; 24 h afterward, the cells are treated with LPS (25 ng/mL) or H_2_O for 24 h prior to analysis. L-lactate levels are depicted as a fold change relative to the H_2_O H_2_O control. *n* = 3 independent experiments, 2 replicates per experiment. Graphs shown as bar graph + SEM Tukey’s post hoc significances denoted by (ns *p* > 0.05, * *p* < 0.05, *** *p* < 0.001, **** *p* < 0.0001). All statistics are reported in Appendix A.

**Table 1 ijms-24-08967-t001:** Primer Sequences.

Target Gene	Forward Sequence (5′→3′)	Reverse Sequence (5′→3′)
** *Hprt* **	CAGTACAGCCCCAAAATGGTTA	AGTCTGGCCTGTATCCAACA
** *Il1b* **	TGGCAACTGTTCCTGAACTCA	GGGTCCGTCAACTTCAAAGAAC
** *Il6* **	CGATGATGCACTTGCAGAAA	ACTCCAGAAGACCAGAGGAA
** *Cxcl16* **	ATCAGGTTCCAGTTGCAGTC	TTCCCATGACCAGTTCCAC
** *Il10* **	ACAAAGGACCAGCTGGACAA	TAAGGCTTGGCAACCCAAGTA
** *Tnfa* **	GGGTGATCGGTCCCCAAA	TGAGGGTCTGGGCCATAGAA

## Data Availability

All raw data are available upon request to the corresponding author.

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
