# Peer review of "Modelling Microglial Innate Immune Memory In Vitro: Understanding the Role of Aerobic Glycolysis in Innate Immune Memory"

_ijms, 2023, doi:10.3390/ijms24108967_

Round 1

Reviewer 1 Report

The authors analyzed an important subject – the correlation between metabolic reprogramming and innate immune memory in microglia, which is related to the Developmental Origins of Health and Disease theory. They developed a model for studying innate immune memory in microglia in vitro. The study is well-designed and performed, and the conclusions are supported by the results.

However, some information are still missing in the study, and therefore there is a possibility to improve the manuscript:

1.     Regarding cytokines analyzed, the authors based their conclusions only on the RT-qPCR results, which is also related to the study design which is limited to the 24-hour time interval. However, the changes in the protein level would be a stronger proof of concept.

2.     The authors performed one-, two-, and three-way ANOVA, followed by post hoc comparisons by Tukey’s test. However, the information on equality of variance is missing. Please explain how did you check the homogeneity of variance?

Minor suggestion:

#Fig 3. LPS but not BAFF induces a strong cytokine response through aerobic metabolism

 Comment: This claim is only a speculation, it should be reformulated.   

    L134 ...AVOVA

    Comment: This should be corrected to ANOVA.

Author Response

Thank you for taking your time to review our paper and provide your comments and concerns. First, we would like to thank you for your positive comments regarding the study design and the importance of the work. Second, we have addressed each of your concerns, as below:

  1. Regarding cytokines analyzed, the authors based their conclusions only on the RT-qPCR results, which is also related to the study design which is limited to the 24-hour time interval. However, the changes in the protein level would be a stronger proof of concept.

Thank you for this comment - you are right that qPCR levels do not always correlate with the protein levels. However, you are also correct that this is related to the study design and a consequence of performing this on a short time point and as a result of a longer culture paradigm with rapidly dividing cells. We have added this as a limitation in the discussion and added the importance of considering protein levels in vivo models (see lines:487-493)

  1.     The authors performed one-, two-, and three-way ANOVA, followed by post hoc comparisons by Tukey’s test. However, the information on equality of variance is missing. Please explain how did you check the homogeneity of variance?

Thank you for the comment. You are correct that ANOVAs both assume equal variance and normalcy to make comparisons. To ensure we are not making false comparisons, we test for equal variance using the Spearman's test for heteroscedasticity or the Brown-Forsythe Test. Likewise, we test for normalcy using the Shapiro-Wilk test. Please see the methods section 4.6 for the correction.
#Fig 3. LPS but not BAFF induces a strong cytokine response through aerobic metabolism
Comment: This claim is only a speculation, it should be reformulated.  
You are correct that this title is speculatory and misleading - please see Figure 3 for the new title

“LPS but not BAFF induces rapid induction of lactate production”
L134 ...AVOVA
Comment: This should be corrected to ANOVA.
Thank you for pointing out the typo - it is now fixed!

Again, thank you very much for taking the time to review our paper and for your kind words.

Reviewer 2 Report

Microglia cells play an extremely important role in aging and disease.  Microglia have two main memory states, training and tolerance, which are associated with increased or attenuated expression of inflammatory cytokines respectively. However, the mechanisms differentiating these two distinct states are not well understood. In this work, the authors investigated mechanisms underlying training versus tolerance memory paradigms in vitro in BV2 cells using B-cell activating factor (BAFF) or bacterial lipopolysaccaride (LPS) as a priming stimuli followed by lipopolysaccaride as a second hit. BAFF followed by LPS showed enhanced responses indicative of priming whereas LPS then LPS again for the 2nd stimuli caused tolerance and a reduced response. The main difference between the BAFF versus the LPS stimuli was the induction of aerobic glycolysis. Inhibiting aerobic glycolysis during the priming stimulus using sodium oxamate prevented the establishment of the tolerized memory state. In addition tolerized microglia were unable to induce aerobic glycolysis upon LPS restimulus. Therefore, we conclude that aerobic glycolysis triggered by the first LPS stimulus was a critical step in the induction of innate immune tolerance.  The results are very interesting. However, are there any study in vivo studies supporting the results?

Author Response

Thank you for taking your time to read and review our research. We really appreciate your summary and saying that our results are interesting! With regards to the in vivo comment, there is a body of research which examines the interplay between microglia metabolism and function more closely however this has yet to be examined clearly in the context of immune memory. Therefore, this is a future direction for our research, however, as mentioned in the discussion transient pharmacological inhibitions such as what we assess in vitro are not easy to examine in vivo as genetic manipulation would not be transient and pharmacological manipulation would not be microglia specific. Other studies have suggested the role of metabolism in microglia immune tolerance through differential metabolic genes in RNA sequencing (Hayes, 2022), however, this mechanism is very poorly understood and needs further exploration. In addition, when working in vivo using adult immune memory models such as the 4X LPS model published by Wendeln (2018), we found that it was difficult to examine the impact of a first immune stimulus when 3 injections were required to induce a tolerance phenotype. Therefore, we are working on developing a heterologous single injection first hit paradigm for training and tolerance in vivo which we will use to examine and manipulate metabolic changes. We have added a concise version of this to the discussion see lines 507-512. Again, thank you for your comments and taking your time to review our research. 

Round 2

Reviewer 1 Report

The authors replied satisfactorily to all the raised issues and improved the manuscript.

Reviewer 2 Report

The manuscript has been improved and can be published in its present form

English is good; can be nevertheless improved by careful editing